# PPP2R5D-Related Intellectual Disability and Neurodevelopmental Delay: A Review of the Current Understanding of the Genetics and Biochemical Basis of the Disorder

**DOI:** 10.3390/ijms21041286

**Published:** 2020-02-14

**Authors:** Dayita Biswas, Whitney Cary, Jan A. Nolta

**Affiliations:** 1SPARK Program Scholar, Institute for Regenerative Cures, University of California, Sacramento, CA 95817, USA; biswasd3779@student.sanjuan.edu; 2Stem Cell Program, UC Davis School of Medicine. The University of California, Sacramento, CA 95817, USA; 3UC Davis Gene Therapy Program, University of California, Sacramento, CA 95817, USA

**Keywords:** autism spectrum disorders, PPP2R5D, seizures, neurodevelopmental disabilities, phosphatase, review, intellectual disability, PP2A

## Abstract

Protein Phosphatase 2 Regulatory Subunit B′ Delta (*PPP2R5D*)-related intellectual disability (ID) and neurodevelopmental delay results from germline de novo mutations in the *PPP2R5D* gene. This gene encodes the protein PPP2R5D (also known as the B56 delta subunit), which is an isoform of the subunit family B56 of the enzyme serine/threonine-protein phosphatase 2A (PP2A). Clinical signs include intellectual disability (ID); autism spectrum disorder (ASD); epilepsy; speech problems; behavioral challenges; and ophthalmologic, skeletal, endocrine, cardiac, and genital malformations. The association of defective PP2A activity in the brain with a wide range of severity of ID, along with its role in ASD, Alzheimer’s disease, and Parkinson’s-like symptoms, have recently generated the impetus for further research into mutations within this gene. PP2A, together with protein phosphatase 1 (PP1), accounts for more than 90% of all phospho-serine/threonine dephosphorylations in different tissues. The specificity for a wide variety of substrates is determined through nearly 100 different PP2A holoenzymes that are formed by at least 23 types of regulatory B subunits, and two isoforms each of the catalytic subunit C and the structural subunit A. In the mammalian brain, PP2A-mediated protein dephosphorylation plays an important role in learning and memory. The PPP2R5D subunit is highly expressed in the brain and the PPP2A–PPP2R5D holoenzyme plays an important role in maintaining neurons and regulating neuronal signaling. From 2015 to 2017, 25 individuals with *PPP2R5D*-related developmental disorder were diagnosed. Since then, Whole-Exome Sequencing (WES) has helped to identify more unrelated individuals clinically diagnosed with a neurodevelopmental disorder with pathological variants of *PPP2R5D*. In this review, we discuss the current understanding of the clinical and genetic aspects of the disorder in the context of the known functions of the PP2A–PPP2R5D holoenzyme in the brain, as well as the pathogenic mutations in *PPP2R5D* that lead to deficient PP2A–PPP2R5D dephosphorylation and their implications during development and in the etiology of autism, Parkinson’s disease, Alzheimer’s disease, and so forth. In the future, tools such as transgenic animals carrying pathogenic PPP2R5D mutations, and patient-derived induced pluripotent stem cell lines need to be developed in order to fully understand the effects of these mutations on different neural cell types.

## 1. Background

Protein Phosphatase 2 Regulatory Subunit B′ Delta (*PPP2R5D*)-related intellectual disability (ID) and neurodevelopmental delay is a disorder that mainly occurs due to de novo mutations in the *PPP2R5D* gene [1,2,3,4]. The *PPP2R5D* gene codes for one of the isoforms of the regulatory subunit family B56, of the heteromeric enzyme protein phosphatase 2A (PP2A) [5]. Mutations in *PPP2R5D* are associated with neurodevelopmental delay, autism spectrum disorder (ASD), ID, behavioral challenges, and so forth, which are seen soon after birth [1,2,3,4,6]. In the United States, this disorder is currently referred to as Jordan’s syndrome, named after Jordan Lang, the first child diagnosed in the United States [7]. The severity of *PPP2R5D*-mutation-related ID along with the association of PP2A dysfunction with Alzheimer’s disease [8], Parkinson’s-like symptoms [9,10], and cancer [11,12] has generated a strong need to study the effects of these mutations on the development and functions of different neural cells.

PP2A is a major serine (Ser)/threonine (Thr) phosphatase family comprising more than 100 holoenzymes (reviewed in [13]) that performs diverse and substrate-specific physiologic functions in different tissues (reviewed in [14]). The specificity for a wide variety of tissue-specific substrates is determined through approximately 100 different combinations of the holoenzymes formed by the two A and C isoforms each, along with 23 different isoforms of the B subunits (reviewed in [8,14,15]). PP2A is essential for key cellular processes occurring both during development and in adults, including gene transcription, cell division and growth, and muscle contraction (reviewed in [11,13,14,16,17,18,19,20,21]). De novo mutations within *PPP2R5D* result in defective PP2A–PPP2R5D holoenzyme activities, such as the inability to recognize the substrate or relocate to the nucleus, and deficient catalytic activity [1,3]. The PPP2R5D protein is highly expressed in the human brain [22], where the PP2A–PPP2R5D holoenzyme is involved in controlling the activity of signaling pathways that play roles in the maintenance and functioning of neurons (discussed in detail later).

In this review, we report the current understanding of the structure and functions of the PP2A–PPP2R5D holoenzyme, the pathogenic mutations in *PPP2R5D*, and the associated clinical signs of ID and developmental delay. We have also reviewed the current literature on the role of PPP2A–PPP2R5D in different signaling pathways that are relevant in autism spectrum disorder, Alzheimer’s disease 8, and Parkinson’s disease.

## 2. Genetics

The *PPP2R5D* gene locus is on chromosome 6 in humans and on chromosome 17 in mice [23]. *PPP2R5D* encodes the 602-amino-acid protein PPP2R5D (also known as the B56δ subunit) [24]. Both male and female offspring can have the germline pathogenic mutations [6]. In humans, the majority of the pathogenic mutations in this gene are often missense mutations arising de novo during gestation [3] or may be generated in the family for the first time because of a new variant in germ cells of any one of the parents (or both). In 1996, Del-Mazo’s lab had isolated and cloned a gene initially called Tex271 by differential screening of a subtractive library derived from mouse testis tissue [2]. Subsequently, Tex271 was shown to be the mouse *Ppp2r5d* [2,15].

## 3. Clinical Signs

The clinical signs of *PPP2R5D*-related disorder include moderate to severe intellectual disability, autism spectrum disorder, and neurodevelopmental delay immediately after birth [1,2,3,6]. All patients have a speech impairment [1,2,3,6], hypotonia [1,2,3,6], macrocephaly 2, and frontal bossing (protruding forehead) [1,2,3,6]. Others have susceptibility to complex partial, generalized, or multifocal epilepsy (epileptic spasms or tonic–clonic); global developmental delay; mild orbital hypertelorism (increased distance between the eyes); delay in gross motor skills; and downslanting palpebral fissures [1,2,3,4,6]. Other clinical signs associated with these mutations include ophthalmologic, skeletal, endocrine, cardiac, and genital abnormalities [1,2,3]. The age of affected individuals ranges from 22 months to 53 years [1]. In terms of brain magnetic resonance imaging (MRI) results, all of these individuals present with megalencephaly [1,3,6] and one or more nonspecific MRI findings [1,3,6], such as abnormalities in the white matter, presence of cavum septum pellucidum et vergae and cavum septum pellucidum, small or dysplastic corpus callosum, hydrocephalus, and mild to moderate ventricular dilations [1,3,6]. Because *PPP2R5D*-related clinical signs can overlap with clinical features of other neurodevelopmental disorders, a differential diagnosis is achieved by combining MRI data along with genetic testing of the proband for pathogenic *PPP2R5D* variants [6].

Deletions of the chromosome locus 6p21.1 that contains *PPP2R5D* have been reported in the DECIPHER (DatabasE of genomiC varIation and Phenotype in Humans using Ensembl Resources) databases as being associated with disease [25]. Two patients with chromosome deletions including all of *PPP2R5D* were reported with similar phenotypes [25]. Clinical features of the patients included developmental delay, low-set ears, prominent forehead, and a depressed nasal bridge; a third child had intellectual disability, hypotonia, downslanting palpebral fissures, delayed speech and language development, hydrocephalus, and hypotonia [25]. Another child with gene duplication including part of *PPP2R5D* has been reported with signs of developmental delay [25].

## 4. Etiology

PPP2R5D-related disorder is an autosomal dominant condition since one copy of the mutated gene in each cell is sufficient to cause the disorder. From 2015 to 2017, 25 individuals with *PPP2R5D*-related disorders were reported in the published literature [1,2,3,4]. All patients were diagnosed as having *PPP2R5D*-related ID and neurodevelopmental disorder after the identification of a heterozygous pathogenic variant in *PPP2R5D* on molecular genetic testing established in a proband. Since then, Whole-Exome Sequencing (WES) has helped to identify 100 unrelated individuals clinically diagnosed with a neurodevelopmental disorder with 13 variations of the PPP2R5D mutation [7]. The availability of WES as a part of a newborn genetic disease screening tool is expected to identify many more patients in the future.

## 5. Tissue Expressions and Functions of the PPP2R5D Subunit

Both the *PPP2R5D* gene and the protein are expressed in most organs during embryonic development in humans [22]. After birth, it is expressed in many tissues, but the highest expression is seen in the brain 22. In mice, Ppp2r5d is widely expressed in the embryo [2,26] and postnatally in many tissues, with the highest expression in the brain and spinal cord [2,5,27]. Within the mouse brain, the highest Ppp2r5d protein expression is seen in the striatum and thalamus; intermediate expression in the brain stem, CA1, CA2, CA3 regions, and dentate gyrus of the hippocampus; and relatively weak expression is seen in the cortex and cerebellum [28]. In mammalian somatic cells, PP2A protein has been detected in the cell membrane, cytosol, and nucleus [5,26].

The differential localization of PP2A is determined largely in part by the particular isoforms of the B subunit. The PPP2R5D, PPP2R5γ1, and PPP2Rγ3 isoform containing enzymes were detected in the nucleus, while PP2A containing other isoforms (PPP2R5α, PPP2R5β, and PPP2R5ε) were detected in the cytoplasm [5]. The binding of the PPP2R5D subunit imparts substrate selectivity, intracellular targeting, and catalytic activity [18]. Both PPP2R5D and another isoform (PPP2R5γ3) share a potential bipartite nuclear localization signal starting at residue 547 of PPP2R5D responsible for the targeting of holoenzyme to the nucleus [5,24]. The PP2A–PPP2R5D holoenzyme is also critically involved in the negative regulation of cell growth, remodeling of the chromatin, regulation of transcription, and is a component of the phosphoinositide-3-kinase–protein-kinase-B (PI3K/AKT) growth regulatory cascade [2].

## 6. Pathogenic Mutations in PPP2R5D Leading to Defective PP2A–PPP2R5D Activity

The various single nucleotide substitution mutations in *PPP2R5D* and the associated clinical phenotypes are listed in Table 1. Mutations that resulted in changes in the amino acid code and replacement of highly conserved amino acids in the PPP2R5D protein have been found to be strongly associated with neurodevelopmental delay and intellectual disabilities [1,2,3,6]. In human chromosome 6, four site-specific guanine (G)  >  adenine (A) substitutions in *PPP2R5D* lead to amino acid substitutions glutamic acid (Glu)420lysine (Lys), Glu200Lys, Glu198Lys, and Glu197Lys, and site-specific substitutions of cytosine (C) > thymidine (T), C > G, and C > adenosine (A) lead to proline (Pro)53serine (Ser), Pro201arginine (Arg), and tryptophan (Trp)207Arg substitutions, respectively [1]. All but one *PPP2R5D* mutation are in a highly conserved acidic loop of the PPP2R5D subunit critical for PP2A–PPP2R5D holoenzyme formation [1]. Notably, all the mutations resulted in the introduction of a positively charged arginine or lysine residue. The only exception is the Pro53Ser mutation. Pro53Ser substitution at the N-terminal domain of PPP2R5D changes the binding capacity of the PP2A–PPP2R5D holoenzyme with substrates [1].

The severity of the clinical signs of ID and neurodevelopmental delay appears to be correlated with the degree of the biochemical defect in PP2A activity (Table 1). For example, a severe form of ID has been seen with Glu198Lys and Glu420Lys mutations (Table 1), whereas a relatively milder ID was observed in patients with the Glu197Lys and Glu200Lys mutations [1,2,3,6]. Mechanistically, both Glu200Lys and Glu198Lys substitutions (replacement of negatively charged residues by positively charged residues) impair the subunit-A–subunit-C binding and cause defective PP2A–PPP2R5D-dependent dephosphorylation [1]. However, the milder phenotype suggested that inside the acidic groove formed by the three amino acid residues Glu197, 198, and 200, Glu198 is absolutely critical for the three subunits to interact and bind, while substitutions at Glu200 and Glu197 may not severely inhibit inter-subunit interaction and binding [1,3]. The Glu420 residue is positioned in close proximity to an active core of the catalytic subunit C and is on the outer surface of the PP2A–PPP2R5D holoenzyme complex [1,3]. The reversal of charge (negative to positive) due to Glu420Lys substitution at this site profoundly changes the electrostatic profile of the site that is in close proximity to the active site of the enzyme. This indicates that either disrupting the formation of the holoenzyme or failure of recognition of the substrate has similar effects on defective dephosphorylation.

In vitro, functional expression studies with the human embryonic kidney cell line 293 (HEK293 cells) were done to explore if the subunit interactions were disrupted by the *PPP2R5D* missense mutations [1]. HEK293 cells were transfected with green-fluorescent-protein-tagged wild-type or mutant PPP2R5D subunits and were assayed for subunit–subunit interactions. All ID-associated PPP2R5D variants showed deficient holoenzyme formation (i.e., A- or C-to-PPP2R5D association consistent with a dominant-negative effect) [1]. The only exception was Pro53Ser, as expected [1].

## 7. Association of PP2A–PPP2R5D Dysregulation with Overgrowth Syndrome and Associated Intellectual Disability

Overgrowth syndromes are referred to as a group of disorders that are mainly characterized by excessive prenatal and postnatal growth compared with an age-and-sex-matched control group [2]. Although for most individuals, the direct genetic link to their disorder remains unresolved, several overgrowth-syndrome-associated genes have been reported recently [19]. Interestingly, many of these genes code for the various proteins of the phosphatidylinositol-4,5-bisphosphate 3-kinase/v-AKT murine thymoma viral oncogene homolog 1 (AKT) growth regulatory pathways [19]. Multiple studies have shown evidence that the PP2A–PPP2R5D holoenzyme shows specificity for AKT [18]. Specifically, in certain subcellular compartments, PP2A-B56 can directly dephosphorylate AKT Thr-308 and Ser-473 and inhibit AKT activity [27,29]. Since AKT dephosphorylation promotes cellular proliferation and growth, it is thought to be a negative regulator of the PI3K/AKT-mediated growth regulatory cascade. Consequently, aberrant activation of the PI3K/Akt pathway is primarily caused by loss of function of all negative controllers, including PP2A [30].

Detection of de novo gene mutations by exome sequencing has been highly successful in identifying novel genetic causes of overgrowth syndromes [31,32]. Through this method, overgrowth-syndrome-affected patients (from unrelated families) were identified with de novo mutations in the PP2A subunit B family genes, including *PPP2R5D* 2. This was also supported by the evidence that de novo mutations in AKT1 cause Proteus syndrome (MIM 176920), a disorder of asymmetric overgrowth and tissue hyperplasia affecting many organs [33].

## 8. Animal Models

Transgenic mice in which the *PPP2R5D* gene was knocked out globally (*PPP2R5D*-KO) have been generated and reported [34]. Unexpectedly, these mice were found to be viable and fertile, given the high expression of PP2A–PPP2R5D protein in the embryo of wild-type mice [34] and the fact that the PP2A–PPP2R5D holoenzyme is shown to be a negative regulator of cell division cycle 25C (Cdc25C) phosphatase, which is a key regulator of exit from mitosis during embryonic development [17]. Specifically, PP2A–PPP2R5D dephosphorylates the Cdc25C Thr138 residue [17], and how this process still occurred in the PPP2R5D-KO mice is not clear given that no facial malformations or gross growth abnormalities were observed. It was proposed that a functional compensatory mechanism exists for the Cdc25 dephosphorylation since the embryonic fibroblasts derived from the PPP2R5D-KO mice appear to upregulate the expression of Wee1 kinase (Wee1) protein, the Cdc25-opposing kinase [35]. In the KO mice, no expression of PPP2R5D was seen in the brain and spinal cord as expected, while in the wild-type central nervous system (CNS), the PPP2R5D immunoreactivity was seen partially colocalized with F-actin, microtubules, and tau protein in the neurons. Further, in the KO mice, progressive tau hyperphosphorylation along with marked immunoreactivity for the MC-1-positive tau pre-neurofibrillary tangle conformation (found in AD brain) was seen in the brain stem and the dorsal horn of the cervical spinal cord, but neurofibrillary tangles were absent [35]. Some defects in learning and memory were also seen in the KOs. The same group also reported that *PPP2R5D*-KO mice spontaneously developed hepatic carcinoma via activation of master regulator of cell cycle entry and proliferative metabolism gene (c-Myc) [12]. Transgenic mice expressing different pathogenic *PPP2R5D* mutations are being developed in several laboratories (personal communication).

## 9. Structure of the PP2A Enzyme Complex

The PP2A enzyme complex is composed of three subunits: a 65 kDa structural (scaffolding) subunit A (also known as PR65), a variable/regulatory subunit B, and a 36 kDa catalytic subunit C [16]. The A subunit tightly associates with the C subunit and provides a scaffolding for binding of subunit B [36]. Subunit A contains 15 tandem repeats of a 39-residue sequence known as a huntingtin-elongation-A subunit-TOR motif (HEAT). The motif is organized into an elongated L-shaped molecule [37,38,39]. The catalytic subunit C recognizes the HEAT subunit by specific interaction with the conserved sequences of the tandem repeats 11–15, [40,41,42]. The formation of the PP2A core enzyme causes the HEAT repeats 12–15 to significantly bend towards the amino terminus of the scaffolding subunit A [42]. This conformational rearrangement is critical for its catalytic activity. In eukaryotic cells, the structural subunit A has two isoforms (α and β), which share 86% sequence identity and are ubiquitously expressed [37]. The regulatory B subunits, on the other hand, have highly variable sequences, and the expression levels of various regulatory subunits are highly variable in different tissues [14,16]. In order to bind with specific substrates and catalyze the reaction, the PP2A core enzyme interacts with the various isoforms of the subunit B and forms a holoenzyme. The B subunit contains four subfamilies known as the B (PR55), B′ (B56 or PR61), B″ (PR72), and B″′ (PR93/PR110) subunits, together with approximately 16 known members [14,16]. PPP2R5D is the delta isoform of B56.

## 10. Known Functions of the PP2A–PPP2R5D Holoenzyme 

### 10.1. Targeting of PP2A to the Nucleus

PPP2R5D (and PPP2R5γ) target the PP2A holoenzyme to the nucleus [5]. However, the localization of PP2A–PPP2R5D alters during different phases of the cell cycle. During the interphase, it is present both in the nucleus and the cytoplasm, but in cells undergoing meiosis or that have recently exited from mitosis, the enzyme is only detected in the nucleus [5]. The PPP2R5D protein has been detected and isolated from human erythrocytes, suggesting a non-nuclear localization as well [5].

It has also been shown that PPP2A–PPP2R5D plays a key role in nuclear targeting of the calcium channel, voltage-dependent, beta 4 subunit (CACNB4), which is an auxiliary subunit associated with the pore-forming subunit voltage-gated calcium channels (VGCCs). VGCCs are heteromeric complexes that generate calcium influx in response to electric signals [43], thereby regulating gene expression, synaptic vesicle exocytosis, and neuronal excitability [44,45]. The cytoplasmic CACNB4 controls the level of VGCC expression at the plasma membrane and its biophysical properties [46]. In humans, a mutation of CACNB4, leading to truncation of CACNB4 C-terminus, has been associated with juvenile myoclonic epilepsy [47]. Mechanistic studies done on hippocampal neuronal cultures showed that the epilepsy-causing mutation in the CACNB4 protein resulted in its inability to bind to PPP2R5D and thus prevented translocation of the CACNB4–PP2A–PPP2R5D complex to the nucleus [48]. Nuclear translocation of CACNB4 complexleads to transcription of several genes, including tyrosine hydroxylase, which is noteworthy due to its link to epilepsy [48]. It can be speculated here that pathogenic mutations in PPP2R5D leading to defective PP2A–PPP2R5D activity could have the same effect on the nuclear targeting of CACNB4 and contribute to the development of epilepsy in these patients.

### 10.2. The Role of PPP2A–PPP2R5D in Striatal Dopaminergic Neurotransmission

Dopamine- and cyclic adenosine monophosphate (cAMP)-regulated phosphoprotein (DARPP-32) is a major phosphoprotein with selective expressions at dopaminergic nerve terminals [49]. The phosphorylation state of DARPP-32 involves a mechanism for integrating information arriving at dopaminoceptive neurons via different neurotransmitters, steroid hormones, neuropeptides, and neuromodulators [49]. In striatal neurons, the function of the PP2A–PPP2R5D holoenzyme is to regulate dopaminergic neurotransmission via dephosphorylation of Thr-75 of DARPP-32 [9]. PPP2R5D is first phosphorylated at Ser-566 by a cAMP/protein kinase A (PKA)-dependent pathway [9]. PPP2R5D phosphorylation by PKA results in an increase in the activity of PP2A–PPP2R5D [50]. In vivo studies with mouse striatal brain slices showed that PKA-mediated phosphorylation of PPP2R5D occurs in response to the activation of dopamine-1 (D1) receptors [9]. Once Ser-566 is phosphorylated in PPP2R5D, PP2A–PPP2R5D can dephosphorylate Thr-75, the cyclin-dependent kinase 5 (CDK5) site in DARPP-32 9.

### 10.3. Role of PP2A–PPP2R5D in Neurotrophic Signaling

Nerve growth factor (NGF) is critical for the promotion of growth and survival of the sensory and sympathetic neurons in mammals, including humans (reviewed in [51]). The physiological action of NGF is mediated by two NGF receptors: the tropomyosin-related kinase A (TrkA), which is a tyrosine kinase, and the transmembrane glycoprotein, a pan-neurotrophin receptor p75NTR that regulates signaling through trkA [52]. The PP2A–PPP2R5D holoenzyme has been shown to complex with TrkA in order to dephosphorylate the Ser/Thr residues of NGF to potentiate its tyrosine kinase activity [20]. Therefore, PP2A–PPP2R5D acts at the receptor level to enhance NGF signaling. In in vitro cultures of PC12 cells, PP2A–PPP2R5D has been shown to enhance NGF signaling through the Akt and Ras-mitogen-activated protein kinase cascades, and it promotes neurogenesis and differentiation [20]. Therefore, PP2A–PPP2R5D maintains neurotrophin-mediated developmental and survival signaling through the dephosphorylation of inhibitory serine/threonine residues on the TrkA receptor tyrosine kinase [20].

### 10.4. Role of PP2A–PPP2R5D in Tau Phosphorylation

Yu et al., 2014 [8] were the first to characterize the regulatory B-subunit-specific regulation of tau protein phosphorylation. Specifically, the PPP2R5D and PPP2R3A isoforms of the B subunits are involved in the dephosphorylation of the CDK5 substrate proteins. This indicates that these subunits play a role in the tau dephosphorylation signaling process. Transgenic mice, in which the PPP2R5D gene was knocked out, exhibited spatially restricted tauopathy by deregulation of CDK5 and activation of glycogen synthase kinase-3β (GSK3β) [34]. GSK3β is a tau kinase [34] and its activation results in tau phosphorylation. The PP2A–PPP2R5D complex also leads to tau phosphorylation indirectly. PPP2R5D dephosphorylates protein kinase B (Akt) at the positions Thr-308 and Ser-473 [34]. This results in the inhibition of Akt activity and prevents Akt from phosphorylating GSK3β at position Ser-9. This allows GSK3β to be activated [34]. Additional studies have shown that downregulation of the PPP2R5D subunit expression also corresponds to decreased tau phosphorylation at the amino acid positions Ser-202, Thr-205, Thr-231, and Ser-422, which again supports the notion that the PP2A–PPP2R5D complex can activate GSK3β [8]. Another PP2A B subunit, PPP2R2A, is thought to be involved in reducing the phosphorylation of the tau protein. Thus, PPP2R2A and PPP2R5D are proposed to work in opposite directions regarding tau phosphorylation homeostasis [8]. In an AD model cell line (H4-SWE), PPP2R5D-silencing RNA treatment reduced the level of phosphorylation at Thr-231, Ser-202/Thr-205, and Ser-422, indicating that the PP2A–PPP2R5D holoenzyme activates a few tau-specific kinases. The presence of the hyperphosphorylated isoforms of tau in the neurofibrillary tangles is a histopathological hallmark of Alzheimer’s disease (reviewed in [53]).

Transgenic mice that expressed a dominant-negative mutant form of the catalytic subunit C of PP2A in neurons showed endogenous hyperphosphorylation at two distinct epitopes: the physiological site Ser-202/Thr-205 and the AD-related pathological site Ser-422 [54]. This study also found activation of the extracellular signal-regulated kinase (ERK) and c-Jun N-terminal kinase (JNK) signaling pathway in these mice, suggesting an additional, indirect role of defective PP2A activity in tau hyperphosphorylation, besides the direct dephosphorylation of tau by PP2A. Since PPP2R5D is predominantly expressed in mouse neurons [34], the role of the PP2A–PPP2R5D holoenzyme can be predicted in the tau hyperphosphorylation seen in these animals.

### 10.5. Role of the PP2A–PPP2R5D Holoenzyme in the Progression of Cell Cycle

During mitosis, the genetic material is divided equally between two resulting daughter cells in a tightly regulated series of molecular events. In order for the cells to initiate mitosis, the cyclin-dependent kinase Cdk1 is first activated [55,56]. The active form of the cyclin-dependent kinases, in general, phosphorylate their substrates by transferring the phosphate groups from ATP to specific amino acid residues in the substrates. However, during the interphase, the activity of Cdk1 is repressed by inhibitory phosphorylation of the threonine-14 and tyrosine-15 residues [55,56]. This is catalyzed by the actions of the two inhibitory kinases, namely, Wee1 and the Myt1 kinases [55,56]. During the G2/M transition, the active form of the phosphatase Cdc25C removes these phosphoryl groups, thus activating Cdk1/cyclin B again [57]. The exit from mitosis requires the inactivation of Cdk1/cyclin B initiated by the ubiquitin ligase anaphase-promoting complex/cyclosome (APC/C) and its coactivator cell division cycle 20 (CDC20), which causes proteasomal degradation of cyclin B [58]. Failure to inactivate the Cdk1/cyclin B and degrade cyclin B results in the arrest of the metaphase [59,60]. PP2A seems to plays an important role in mitosis because Cdk1/cyclin B inactivation has been shown to depend on an okadaic-acid-sensitive phosphatase (during the metaphase–anaphase transition) [61]. Okadaic acid is shown to be a potent inhibitor of the serine/threonine protein phosphatases PP1 and PP2A. Also, the PP2A–PPP2R5D holoenzyme complex is a negative regulator of Cdc25C during the interphase. It dephosphorylates Cdc25C threonine/threonine-130 and allows cytosolic sequestration of Cdc25C [17].

## 11. Future Research

Different neuronal cell types are currently being derived by our team and others from induced pluripotent stem cells (iPSC) generated from individuals diagnosed with PPP2R5D-related developmental delay and ID. These neuronal cell phenotypes would have to be characterized, and any aberrant phosphorylation needs to be characterized. These cells will then be used as a platform for screening small molecules or known drugs.

Transgenic mice carrying pathogenic mutations in *PPP2R5D* are currently being developed. These models can be used to investigate tissue-specific defects in dephosphorylation activity in vivo and to screen for potential drugs and gene therapies. The 3D conformation (structure) of PP2A in complex with PPP2R5D-mutated subunits should be generated in order to map protein–protein interactions. Additionally, the genes PPP2R1A and PPP2R5C, which are closely related to PP2R5D, should also be investigated in detail since they are involved in common pathways. A few individuals with mutations in these genes have been diagnosed; however, with WES testing, more will be detected.

The PP2As are the major serine/threonine phosphatases in all tissues. Thus, the specific substrates for the isoforms still need to be characterized in vivo, particularly in the brain. Isoform-specific antibodies specifically against the PP2A–PPP2R5D isoform as well as isoform-specific small-molecule inhibitors/activators and bioassays need to be developed. These tools will facilitate the progress of elucidating the role of PP2A–PPP2R5D and its therapeutic modulations in the context of ID.

## 12. Conclusions

There is currently no cure for ID associated with *PPP2R5D* mutations, but symptoms can be managed, and the treatment of manifestations is the standard treatment for visual impairments, seizures, and developmental delays [6]. Gene therapy approaches are being studied in order to potentially have a future therapeutic effect for this mutation. Since this disorder was only recently discovered and diagnosed correctly, there is much more to research and learn about this rare and debilitating disorder. Other challenges include incorrect diagnosis by professionals as a result of the fact that the PPP2R5D-related neurodevelopmental and intellectual disability has a nonspecific phenotype and can present as different diseases (or solely as autism spectrum disorders). Genetic testing is needed for an accurate diagnosis.

## Figures and Tables

**Table 1 ijms-21-01286-t001:** Clinical features of individuals with different de novo mutations in the Protein Phosphatase 2 Regulatory Subunit B′ Delta (*PPP2R5D*) gene.

PPP2R5D Mutations	Glu197Lys	Glu198Lys	Glu200Lys	Glu420Lys	Pro201Arg	Pro53Ser	Trp207Arg	Reference
Number of individuals diagnosed in the four studies designated as A, B, C, and D	B. 1	A. 6	A. 2	B. 3	A. 1	A. 1	A. 1	A. [1]
		B. 2						B. [3]
		C. 2	B. 1		D. 1	B. 1		C. [4]
		D. 3						D. [27]
Autism spectrum	B. Present	A. Not reported (6/6) B. Present (2/2) C. Present (2/2) D. Not reported	A. Not reported B. Present	B. Present (2/3)	A. Not reported D. Not reported	A. Present B. Present	A. Present	A. [1] B. [3] C. [4] D. [27]
Developmental delay	B. Present	A. Severe (6/6) B. Severe (2/2) C. Moderate (2/2) D. Present (3/3)	A. Mild (2/2) B. Present (2/2)	B. Present (3/3)	A. Not reported D. Present	A. Not reported B. Present	A. Not reported	A. [1] B. [3] C. [4] D. [27]
Unsupported walking	B. 26 months	A. 6–8 years B. 3–8 years C. Not noted D. Not noted	A. 18–30 months B. 28 months	B. 3–5 years (2/3), not walking (1/3)	A. 18 months D. Not reported	A. 5 months to 3 years	A. 1.2 years	A. [1] B. [3] C. [4] D. [27]
Speech	B. ~75 words and can form short sentences	A. Nonverbal (5/6) B. Nonverbal (1/2), and verbal (1/2) C. Not noted D. Not reported	A. Few words B. Few words at 3 years	B. Nonverbal (1/3), a few words (2/3)	A. A few words D. Not reported	A. Few words at 2.5 years (1/1) B. Nonverbal (1/1)	A. Yes, poor intelligibility	A. [1] B. [3] C. [4] D. [27]
Intellectual disability (ID)	B. Moderate	A. Severe (6/6) B. Severe (3/3) C. Moderate (2/2) D. Present (1/3)	A. Present B. Severe	B. Present (2/3)	A. Moderate D. Not reported	A. Severe	A. Moderate	A. [1] B. [3] C. [4] D. [27]
Macrocephaly	B. Present	A. Present (6/6) B. Present (2/2) C. Present (2/2) D. Present (1/3)	B. Absent	B. Present	A. Not present D. Not reported	A. Present	A. Absent	A. [1] B. [3] C. [4] D. [27]
Hypotonia	B. Present	A. Present (6/6) B. Present (2/2) C. Present (2/2) D. present (1/3)	A. Present B. Present	B. Present	A. Present D. Not reported	A. Not reported B. Not reported	A. Present	A. [1] B. [3] C. [4] D. [27]
Number of individuals diagnosed in the four studies designated as A, B, C and D	B. 1	A. 6 B. 2 C. 2 D. 3	A. 2 B. 1	B. 3	A. 1 D. 1	A. 1 B. 1	A. 1	A. [1] B. [3] C. [4] D. [27]
Behavioral Abnormalities	B. Anxiety in new situations	A. Not reported B. Not tested C. Not tested	A. Not tested B. Not tested	B. Aggressive, stereotypies, impulse control issues (2/3)	-	A. Not reported B. Not reported	A. Not reported	A. [6] B. [1] C. [3]
Seizures	B. Not reported	A. Present (2/6) B. Present (1/2) C. Present (2/2)	- B. Present	B. Absent	A. Present (multifocal) D. Not reported	A. Not reported B. Not reported	A. Absent	A. [6] B. [1] C. [3]

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
