# Peer review of "PPP2R5D-Related Intellectual Disability and Neurodevelopmental Delay: A Review of the Current Understanding of the Genetics and Biochemical Basis of the Disorder"

_ijms, 2020, doi:10.3390/ijms21041286_

Round 1

Reviewer 1 Report

In this manuscript, Biswas et al. review the existing literature related to PPP2R5D mutations. Although potentially interesting and important review, I cannot recommend this manuscript for publication due to numerous issues.

In the title, the authors mention PPP2R5D-related neurodevelopmental disability and then go on to talk about PPP2R5D-related intellectual disability. This is not the same (and also, intellectual disability does not relate to neurodevelopmental disorders).

I am afraid that after reading this manuscript, I got the impression that this is an undergrad essay written by a student with not so good understanding of the topic, presented topics, etc. This impression was further strengthened by numerous typos, grammatical mistakes, a narrative that goes back and forth and unacceptable under-referencing.

 Examples:

- many abbreviations were not properly introduced

- “PP2A-PPP2R5D mediated protein dephosphorylation is a regulated process similar to protein 25 phosphorylation in the CNS  - actually opposite” – those two processes are opposite to each other, not similar.

- NGF is not a neutrophil growth factor (reference 31 nicely explains what it is). A neutrophil is a type of a white blood cell, and I believe that the authors talk about nerves and neurotrophic factors.

- “Preclinical research has shown that it may be possible to turn the disorder on and off. Finally, learning how to modify the genes in the PPP2R5D-related intellectual disabilities will have implications for these disorders, which affect many people worldwide.” – this is hardly acceptable for an undergrad essay (e.g. how the disorder can be turned on and off).

- The authors several times mention “malformations” without even giving a hint of what they might be

- autism, autism spectrum, and autism spectrum disorders appear in the text – it is a good practice to use the same terminology

- “PPP2R5D-related intellectual disability (I.D.) is a neurodevelopmental disorder that mainly occurs due to de novo mutations in the PPP2R5D gene. It is associated with various malformations, autism spectrum disorders, intellectual disability (I.D.), seizures, and behavioral challenges. In the United States (U.S.), this condition is currently referred to as Jordan’s syndrome, named after Jordan Lang, the first child diagnosed in the U.S.” – There is not a single reference to support these statements! This is simply not acceptable. And there are many more similar examples throughout the text.

- “PPP2R5D is 65 located on chromosome 17 in mice[5], and on chromosome 23 in cattle” – yet another example of missing references (e.g. cattle). And there are many more similar examples in this manuscript.

- “PPP2R5D has a region at its 66 3′ end that is highly homologous to the 3′ end of the male-enhanced-antigen. They share a region 67 (consisting of 14 base pairs) at the 3′ UTR end but in complementary and reverse orientation [5].” – This comes out of the blue, under-referenced, not explained what is the male enhanced antigen and why it is important to mention it here.

- “P53S ( at the N-terminal domain of PPP2R5D) changes the binding capacity of 152 PP2A- PPP2R5D with substrates or introduces a novel phosphorylation site impairs the regulation 153 by protein kinases…” – due to grammatical issues, this sentence makes no sense (and there are many more similar examples in the manuscript).

- “In general, phosphatases cause proteins to be inactivated, whereas kinases activate [12]” – one more reason why I got the impression that this is an undergrad essay

- “The PP2A holoenzyme with the B subunit PPP2R5D is active dopamine- and cAMP-(cyclic 220 adenosine monophosphate) regulated phosphoprotein (DARPP-32 phosphatase), in the central nervous system (CNS). In this cellular pathway, PPP2R5D is phosphorylated and activated….” – Which cellular pathway? Very unclear.     

- “PP2A308 PPP2R5D knockout mice have been generated and reported [23]. They are viable and fertile. In these mice, the expression of PPP2R5D was knocked down in the brain and spinal cord” – is this a knockout or a knockdown model – cannot be both.

- “The creation of PPP2R5D and other isoform mutated transgenic mice currently in development may be investigated as models for neurodevelopmental disability.” – this sentence makes no sense (e.g. other isoform mutated transgenic mice).

- “Additionally, because the enzyme PP2A 334 is common and present in multiple pathways, it is difficult to establish which cellular pathways are affected by this disease and need to be targeted.” – sure, but it can be done (I am sorry, but I must say that this is simply not a serious writing)

- “However, the presence of a large number of the PP2A isoform indicate that they perform functions that are non-redundant.”   This conclusion is not correct (if anything it is usually the opposite) – can the authors explain their reasoning?

- “Genomic testing is needed for an accurate diagnosis” – why is this? Why not genetic testing is the mutations are known?

my last comment is about acknowledgments. The authors say “Acknowledgments: Through Jordan's Guardian Angels, the foundation established by a patient's family, an international team of clinicians and scientists have been collaborating on research since 2017 to learn about the mutations and their involvement with neurodevelopmental disorders, including autism, intellectual disabilities, behavioral challenges, seizures, and Alzheimer's disease. In 2018, $12 million was allocated by the state of California to study PPP2R5D and its implications specifically” – this is not an acknowledgment but rather an advertisement.

Author Response

Comments and Suggestions for Authors

In this manuscript, Biswas et al. review the existing literature related to PPP2R5D mutations. Although potentially interesting and important review, I cannot recommend this manuscript for publication due to numerous issues.

Reviewer’s comment: In the title, the authors mention PPP2R5D-related neurodevelopmental disability and then go on to talk about PPP2R5D-related intellectual disability. This is not the same (and also, intellectual disability does not relate to neurodevelopmental disorders).

Author’s response: We have changed the title to include both neurodevelopmental and intellectual disability. We have described the clinical features of the PPP2R5D disorder which include signs with a neurodevelopmental delay such as speech and language delay.

I am afraid that after reading this manuscript, I got the impression that this is an undergrad essay written by a student with not so good understanding of the topic, presented topics, etc. This impression was further strengthened by numerous typos, grammatical mistakes, a narrative that goes back and forth and unacceptable under-referencing.

Author’s response: The entire manuscript has been carefully revised and checked for grammatical issues. References have been added and many paragraphs have been rewritten for better clarity. Redundant pieces of information have been removed.

 Examples:

- many abbreviations were not properly introduced

Author’s response: abbreviations have been properly introduced.

- “PP2A-PPP2R5D mediated protein dephosphorylation is a regulated process similar to protein 25 phosphorylation in the CNS  - actually opposite” – those two processes are opposite to each other, not similar.

Author’s response: This information has been corrected.

- NGF is not a neutrophil growth factor (reference 31 nicely explains what it is). A neutrophil is a type of a white blood cell, and I believe that the authors talk about nerves and neurotrophic factors.

Author’s response: The full name of NGF has been corrected

- “Preclinical research has shown that it may be possible to turn the disorder on and off. Finally, learning how to modify the genes in the PPP2R5D-related intellectual disabilities will have implications for these disorders, which affect many people worldwide.” – this is hardly acceptable for an undergrad essay (e.g. how the disorder can be turned on and off).

Author’s response: These lines have been removed.

- The authors several times mention “malformations” without even giving a hint of what they might be.

Author’s response: The specific malformation,’ ophthalmologic, skeletal, endocrine, cardiac, and genital malformations’ have been indicated.

- autism, autism spectrum, and autism spectrum disorders appear in the text – it is a good practice to use the same terminology

Author’s response The term autism spectrum disorder has been used throughout the text.

- “PPP2R5D-related intellectual disability (I.D.) is a neurodevelopmental disorder that mainly occurs due to de novo mutations in the PPP2R5D gene. It is associated with various malformations, autism spectrum disorders, intellectual disability (I.D.), seizures, and behavioral challenges. In the United States (U.S.), this condition is currently referred to as Jordan’s syndrome, named after Jordan Lang, the first child diagnosed in the U.S.” – There is not a single reference to support these statements! This is simply not acceptable. And there are many more similar examples throughout the text.

Author’s response: References have been added to this section.

- “PPP2R5D is 65 located on chromosome 17 in mice[5], and on chromosome 23 in cattle” – yet another example of missing references (e.g. cattle). And there are many more similar examples in this manuscript.

Author’s response: We have deleted the reference to cattle.

- “PPP2R5D has a region at its 66 3′ end that is highly homologous to the 3′ end of the male-enhanced-antigen. They share a region 67 (consisting of 14 base pairs) at the 3′ UTR end but in complementary and reverse orientation [5].” – This comes out of the blue, under-referenced, not explained what is the male enhanced antigen and why it is important to mention it here.

Author’s response: We have deleted this part as we did not find and study that has reported interactions between MEA1 and PPP2R5D.

- “P53S ( at the N-terminal domain of PPP2R5D) changes the binding capacity of 152 PP2A- PPP2R5D with substrates or introduces a novel phosphorylation site impairs the regulation 153 by protein kinases…” – due to grammatical issues, this sentence makes no sense (and there are many more similar examples in the manuscript).

Author’s response: This sentence has been modified (page4).

- “In general, phosphatases cause proteins to be inactivated, whereas kinases activate [12]” – one more reason why I got the impression that this is an undergrad essay

Author’s response: This has been removed.

- “The PP2A holoenzyme with the B subunit PPP2R5D is active dopamine- and cAMP-(cyclic 220 adenosine monophosphate) regulated phosphoprotein (DARPP-32 phosphatase), in the central nervous system (CNS). In this cellular pathway, PPP2R5D is phosphorylated and activated….” – Which cellular pathway? Very unclear.   

  Author’s response: This paragraph has been modified for better clarity

- “PP2A308 PPP2R5D knockout mice have been generated and reported [23]. They are viable and fertile. In these mice, the expression of PPP2R5D was knocked down in the brain and spinal cord” – is this a knockout or a knockdown model – cannot be both.

Author’s response: This is a knockout model. We have corrected the information.

- “The creation of PPP2R5D and other isoform mutated transgenic mice currently in development may be investigated as models for neurodevelopmental disability.” – this sentence makes no sense (e.g. other isoform mutated transgenic mice).

Author’s response: This sentence has been rewritten

- “Additionally, because the enzyme PP2A 334 is common and present in multiple pathways, it is difficult to establish which cellular pathways are affected by this disease and need to be targeted.” – sure, but it can be done (I am sorry, but I must say that this is simply not a serious writing)

Author’s response: This has been taken out.

- “However, the presence of a large number of the PP2A isoform indicate that they perform functions that are non-redundant.”   This conclusion is not correct (if anything it is usually the opposite) – can the authors explain their reasoning?

Author’s response: We have taken out this sentence since there is no data to back it up.

- “Genomic testing is needed for an accurate diagnosis” – why is this? Why not genetic testing is the mutations are known?

Author’s response: We agree with the reviewer and have modified the sentence.

my last comment is about acknowledgments. The authors say “Acknowledgments: Through Jordan's Guardian Angels, the foundation established by a patient's family, an international team of clinicians and scientists have been collaborating on research since 2017 to learn about the mutations and their involvement with neurodevelopmental disorders, including autism, intellectual disabilities, behavioral challenges, seizures, and Alzheimer's disease. In 2018, $12 million was allocated by the state of California to study PPP2R5D and its implications specifically” – this is not an acknowledgment but rather an advertisement.

Author’s response: Since Jan Nolta is funded by Jordan's Guardian Angels, we will keep this in acknowledgment. We have removed rest of the paragraph.

Reviewer 2 Report

This review of Biswas and colleagues provides an interesting and detailed overview of the PPP2R5D-associated neurodevelopmental disorders and of its protein function.

However, some parts of the manuscript should be adapted or supplemented regarding the following comments:

Major comments:

The authors should have the manuscript revised for English grammar and style before resubmission Mutation nomenclature should be modified in accordance to the Human Genetic Variation Society guidelines: protein variants should be written in brackets when RNA nor protein have been analyzed, because they represent predictions. In addition, amino acids should be indicated with the three letters abbreviation References are missing in the last part of the background paragraph (page 2, rows 68-78) The authors mention the chromosomal location of the gene in human, mouse and cattle and disclose the existence of a minimal region of complementarity to the male-enhanced-antigen. It would be rather interesting to know the level of homology/conservation among the different species The total number of individuals with PPP2R5D mutations changes throughout the manuscript: in some parts 23 individuals are mentioned, in other parts 26. The authors should correct this The authors mention twice that “approximately 250000 children and adults might be affected by PPP2R5D mutations”, which seems quite an overestimation given how rare this type of neurodevelopmental disorders is. Where does this estimation come from? Paragraph 2.3 (Pathogenic mutations) is hard to follow due to continue changes in topic. First the structural changes caused by E200K, E197K and E420K are described. Then the possible pathogenic mechanism is discussed, followed by another explanation of the structural effects of two additional substitutions (E197K, P53S). Afterwards, the authors argue about transcript expression. Transcript expression should be moved to paragraph 2.2, where it has been already mentioned. The structural effects of the missense mutations should be discussed all together and the paragraph should be concluded with the possible pathogenic mechanisms Few deletions encompassing the PPP2R5D gene have been reported in Decipher. Given the different pathogenic mechanism proposed, it would be interesting to make a comparison between the phenotype of the deletion patients (loss-of-function) and the phenotype of the patients with missense substitutions (dominant negative) The patients described by Yeung at al. should be included in Table 1

Minor considerations:

In the paper, the authors use the phrase “de novo”, “in vitro” and “in vivo” both with and without italics. They should make it uniform across the paper The meaning of abbreviations like I.D., N.D. or ASD is specified several times throughout the manuscript. The authors should specify the meaning of these abbreviations only once the first time the term is used.

Author Response

Comments and Suggestions for Authors

This review of Biswas and colleagues provides an interesting and detailed overview of the PPP2R5D-associated neurodevelopmental disorders and of its protein function.

However, some parts of the manuscript should be adapted or supplemented regarding the following comments:

Major comments:

1. The authors should have the manuscript revised for English grammar and style before resubmission.

Author Response: The manuscript has been revised for English grammar and style.

2. Mutation nomenclature should be modified in accordance to the Human Genetic Variation Society guidelines: protein variants should be written in brackets when RNA nor protein has been analyzed, because they represent predictions.

Author response: Mutation nomenclature should be modified in accordance to the Human Genetic Variation Society. Protein variants have been written in brackets when appropriate.

3. In addition, amino acids should be indicated with the three letters abbreviation.

Author Response: amino acids have been indicated with the three letters abbreviation.

4. References are missing in the last part of the background paragraph (page 2, rows 68-78).

Author Response: References have been added to the last part of the background paragraph.

5. The authors mention the chromosomal location of the gene in human, mouse and cattle and disclose the existence of a minimal region of complementarity to the male-enhanced-antigen. It would be rather interesting to know the level of homology/conservation among the different species.

Author response: Reference to cattle has been omitted. The homology between human and mouse has been included. We have omitted the refence to MEA1 since there is no interaction between MEA1 and PPP2R5D.

6. The total number of individuals with PPP2R5D mutations changes throughout the manuscript: in some parts 23 individuals are mentioned, in other parts 26. The authors should correct this.

Author response: This has been corrected based on current published literature. Reports of pateints identified through the Jordans Agel wesite, but not published  have been excluded.

7. The authors mention twice that “approximately 250000 children and adults might be affected by PPP2R5D mutations”, which seems quite an overestimation given how rare this type of neurodevelopmental disorders is. Where does this estimation come from?

Author response: We have removed these lines.

8. Paragraph 2.3 (Pathogenic mutations) is hard to follow due to continue changes in topic. First the structural changes caused by E200K, E197K and E420K are described. Then the possible pathogenic mechanism is discussed, followed by another explanation of the structural effects of two additional substitutions (E197K, P53S).

Author response: We have modified this section for better clarity

9. Afterward, the authors argue about transcript expression. Transcript expression should be moved to paragraph 2.2, where it has been already mentioned.

Author response: Transcript expression should be moved to the appropriate section.

10. The structural effects of the missense mutations should be discussed all together and the paragraph should be concluded with the possible pathogenic mechanisms.

Author response: The paragraph has been modified according to reviewer’s suggestions.

11. Few deletions encompassing the PPP2R5D gene have been reported in Decipher. Given the different pathogenic mechanisms proposed, it would be interesting to make a comparison between the phenotype of the deletion patients (loss-of-function) and the phenotype of the patients with missense substitutions (dominant negative).

Author response: Cases with gene duplication and gene deletion containing PP2R5D have been included from DECIPHER.

12. The patients described by Yeung at al. should be included in Table 1

Author response: Patients described by Yeung at al. have been included in Table 1

Minor considerations:

13. In the paper, the authors use the phrase “de novo”, “in virto” and “in vivo” both with and without italics. They should make it uniform across the paper.

Author response: italics have been used throughout the manuscript.

14. The meaning of abbreviations like I.D., N.D. or ASD is specified several times throughout the manuscript. The authors should specify the meaning of these abbreviations only once the first time the term is used.

Author response: Abbreviations have been properly introduced.

Round 2

Reviewer 1 Report

Following revisions, I recommend publication of this manuscript.